# Healthspan Maintenance and Prevention of Parkinson’s-like Phenotypes with Hydroxytyrosol and Oleuropein Aglycone in *C. elegans*

**DOI:** 10.3390/ijms21072588

**Published:** 2020-04-08

**Authors:** Giovanni Brunetti, Gabriele Di Rosa, Maria Scuto, Manuela Leri, Massimo Stefani, Christian Schmitz-Linneweber, Vittorio Calabrese, Nadine Saul

**Affiliations:** 1Department of Biomedical and Biotechnological Sciences, University of Catania, 95125 Catania, Italy; q.burneti@gmail.com (G.B.); dirosagabriele85@gmail.com (G.D.R.); mary-amir@hotmail.it (M.S.); 2Department of Experimental and Clinical Biomedical Sciences “Mario Serio”, University of Florence, Viale Morgagni 50, 50134 Florence, Italy; manuela.leri@unifi.it (M.L.); massimo.stefani@unifi.it (M.S.); 3Department of Neuroscience, Psychology, Area of Medicine and Health of the Child of the University of Florence, Viale Pieraccini, 6 - 50139 Florence, Italy; 4Humboldt University of Berlin, Faculty of Life Sciences, Institute of Biology, Molecular Genetics Group, Philippstr. 13, House 22, 10115 Berlin, Germany; christian.schmitz-linneweber@rz.hu-berlin.de

**Keywords:** *C. elegans*, polyphenols, olive oil, healthspan, lifespan, ageing, Parkinson’s disease

## Abstract

Numerous studies highlighted the beneficial effects of the Mediterranean diet (MD) in maintaining health, especially during ageing. Even neurodegeneration, which is part of the natural ageing process, as well as the foundation of ageing-related neurodegenerative disorders like Alzheimer’s and Parkinson’s disease (PD), was successfully targeted by MD. In this regard, olive oil and its polyphenolic constituents have received increasing attention in the last years. Thus, this study focuses on two main olive oil polyphenols, hydroxytyrosol (HT) and oleuropein aglycone (OLE), and their effects on ageing symptoms with special attention to PD. In order to avoid long-lasting, expensive, and ethically controversial experiments, the established invertebrate model organism *Caenorhabditis elegans* was used to test HT and OLE treatments. Interestingly, both polyphenols were able to increase the survival after heat stress, but only HT could prolong the lifespan in unstressed conditions. Furthermore, in aged worms, HT and OLE caused improvements of locomotive behavior and the attenuation of autofluorescence as a marker for ageing. In addition, by using three different *C. elegans* PD models, HT and OLE were shown i) to enhance locomotion in worms suffering from α-synuclein-expression in muscles or rotenone exposure, ii) to reduce α-synuclein accumulation in muscles cells, and iii) to prevent neurodegeneration in α-synuclein-containing dopaminergic neurons. Hormesis, antioxidative capacities and an activity-boost of the proteasome & phase II detoxifying enzymes are discussed as potential underlying causes for these beneficial effects. Further biological and medical trials are indicated to assess the full potential of HT and OLE and to uncover their mode of action.

## 1. Introduction

Neurodegenerative diseases are becoming increasingly prevalent in the ageing populations of industrialized nations, going hand in hand with the increase in life expectancy. These disorders, which include Alzheimer’s disease and Parkinson’s disease, share a common feature: the accumulation of misfolded proteins in pathological inclusions [1].

Parkinson’s disease (PD) is a chronic, age-related and adult-onset neurodegenerative disorder characterized by the loss of dopaminergic neurons in an area of the midbrain called substantia nigra (SN) along with intraneuronal inclusions known as Lewy bodies, which contain amyloid aggregates of misfolded α-synuclein [2,3,4]. PD is considered today as the most common movement disorder that affects 1–2 per 1000 of the population and since the prevalence is increasing with age, PD affects 1% of the population above 60 years [5]. There are dozens of PD-related symptoms and signs but the most typical are motor deficits including tremors, muscle rigidity, bradykinesia, and impaired gait. Among non-motor symptoms, the most common are olfactory dysfunction, cognitive impairment, psychiatric symptoms, and autonomic dysfunction [6]. PD is a multifactorial disorder and the majority of PD cases are sporadic with unknown aetiology possibly caused by an association of genetic and environmental risk factors. At least 23 loci and 19 disease-causing genes for PD have been identified and designated as PD-causing genes [7]. 

Several hypotheses have been proposed regarding the cause of loss of dopaminergic neurons in PD, whereas oxidative stress, in particular, is strongly associated with the development of PD [8,9]. Other studies have shown defects in the mitochondrial complex-I of neurons, which lead to impaired energy metabolism and cell death [10]. Furthermore, the proteolytic hypothesis describes nigral neuron loss in PD as a result of toxic accumulation of aggregates of misfolded proteins, notably α-synuclein, resulting in neuro-inflammation [11]. Dopaminergic neurons of substantia nigra pars compacta appear particularly vulnerable to the harmful effects of α-synuclein aggregates [12]. Since ageing is a major risk for PD, it has been hypothesized that PD could be, at least in part, a type of segmental ageing, in which the viability of dopaminergic (DA) neurons is impaired by so far unknown localized and accelerated ageing mechanisms [13].

Neurodegenerative disorders are associated with high morbidity and mortality, and few effective options are available for their treatment [1]. Thus, many studies have been conducted focusing on natural compounds present in food as important molecules against neurodegenerative diseases such as PD [14,15,16,17]. Several lines of evidence support the beneficial effect of the Mediterranean diet (MD) in preventing neurodegeneration, possibly due to its richness in polyphenols [18,19]. Natural polyphenols exert numerous biological activities, like antioxidant, anti-inflammatory, antiviral, antibacterial, antiproliferative, and anticarcinogenic capacities (reviewed in Stevanovic, et al. [20]), as well as cellular redox state modulation activities through direct action on enzymes, proteins and receptors [21,22]. In addition, in patients affected by osteoarthritis or cardiovascular diseases, beneficial epigenetic chromatin modifications were also caused by polyphenols [23,24,25].

One possible mode of action of natural polyphenols is the hormesis effect. The biological processes underlying hormetic dose–response, recently, focused attention in the field of neuroprotection, which was mainly elucidated through the exploitation of bioactive polyphenols against the main age-related diseases, particularly in PD [13]. In this light, low levels of exogenous and endogenous stressors have been reported to display hormetic characteristics that induce neurophysiological mechanisms of maintenance and repair, including heat shock, the application of pro-oxidants, as well as the application of polyphenols from plants [26]. Recently, it has been postulated that the MD exerts healthy effects through hormetic mechanisms, as specific olive oil polyphenols (e.g., oleuropein and hydroxytyrosol) likely counteract the effects of neuro-inflammatory stimuli by acting as modulators of stress responsive mechanisms, which result in adaptive stress resistance [27]. Moreover, in vivo studies suggest that a diet rich in phytochemicals may enhance neuroplasticity and stress resistance to neuro-inflammation, mitigating or preventing neurodegenerative changes in the brain that are typical in a number of age-related disorders, including PD [28,29,30].

It has been hypothesized that extra virgin olive oil polyphenols could be among the main determinants of the beneficial effect of the MD [31,32,33]. Extra virgin olive oil (EVOO) contains approximately 36 phenolic compounds [21], which represent the main group of antioxidants found in virgin olive oil. The main phenolic subclasses present in olive oil are phenolic alcohols, phenolic acids, flavonoids, lignans, and secoiridoids [34], whereas the latter represent the largest quantity in the EVOO. The main secoiridoids in olive oil are oleuropein aglycone and ligstroside, which undergo transformation to hydroxytyrosol or tyrosol, respectively through two enzymes (beta glucosidase and esterase) in the gastrointestinal tract [35]. 

In the present paper, we focus on health effects of hydroxytyrosol (HT; Figure 1B) and oleuropein aglycone (OLE; Figure 1A) as neuroprotective agents against PD in the light of recent studies indicating that OLE and HT can be beneficial against PD by stabilizing the monomeric state of α-synuclein, thus, favouring the growth of aggregates devoid of toxicity [36,37]. Furthermore, previous studies have shown that these molecules are strongly protective against neurodegeneration in different transgenic models of Aβ deposition [38,39,40]. Moreover, we have to consider that hydroxytyrosol is a by-product of dopamine oxidative metabolism [41] and in the last years the gold standard therapy against PD has relied on restoring the optimum level of dopamine [42]. 

Tests with mammalian models are very powerful, but are expensive, long-lasting and cause ethical concerns. Therefore, the widely used model organism *Caenorhabditis elegans* (*C. elegans*) was applied for this study. Despite its simple structure, *C. elegans* features several tissues and organs in alignment to higher animals, like muscles, a nervous system, an epidermis, a gastrointestinal tract, and gonads [43]. Furthermore, about 50% of the human protein-coding genome has recognizable worm orthologs [44]. Last but not least, neurological pathways are highly conserved between invertebrates and mammals, and numerous neurodegenerative disease-related transgenic and mutant strains are available in the nematodes [45]. In addition to several neuro-protective substance screenings [45], numerous polyphenols were also successfully tested for their general health and lifespan benefits in *C. elegans* [46,47,48], making this nematode an optimal model to study the neuroprotective effects of olive polyphenols. 

It was hypothesized that the polyphenol OLE and its main metabolite HT increase the mean lifespan of *C. elegans* in the presence and in the absence of stress conditions. Furthermore, it is assumed that they are able to counteract the age-related decline of general health parameters, which were assessed by determining the swim behaviour as a measure of overall body fitness as well as the intestinal autofluorescence, being one of the most popular ageing biomarkers [49]. Furthermore, numerous cell culture studies were already successfully performed to verify the anti-PD effects of olive ingredients, as summarized in Angeloni, Malaguti, Barbalace, and Hrelia [22], however, in vivo studies are hardly present. Therefore, by using a chemically induced and two transgenic PD models of *C. elegans*, the polyphenolic treatments were tested for their anti-PD effects in vivo. Although *C. elegans* is not able to develop PD, the PD models feature characteristic attributes related to PD. Besides the swim performance, neuronal degeneration as well as α-synuclein accumulation were taken into account to assess the anti-PD potential.

## 2. Results

### 2.1. Oleuropein Aglycone and Hydroxytyrosol Extended the Survival of Wild-type *C. elegans* after Heat Stress

To find the optimal concentration for this study, the treatments with polyphenols were initially tested at different concentrations in a heat stress-resistance test. Due to the relatively fast and simple execution, heat stress resistance is frequently measured to screen treatments for potential health and lifespan benefits [50,51,52]. Since stress resistance abilities are strongly correlated to ageing, ageing-associated diseases and lifespan [53,54,55,56], this test was also used as a pre-test in the current study. The monitored survival after stress exposure revealed an improved stress resistance in OLE-treated nematodes (Figure 2A). Two-hundred-and-fifty µg/mL and 500 µg/mL OLE provoked statistically significant changes in all biological repeats performed. The mean lifespan after heat stress increased by 15% and 22% in the OLE 250 µg/mL and OLE 500 µg/mL treated group, respectively. However, the survival differences between OLE 250 µg/mL and OLE 500 µg/mL treated nematodes were not significant. Furthermore, no significant survival benefits were observed with 30 and 100 µg/mL compared to control. Due to the highest percentage benefit, 500 µg/mL OLE was chosen for subsequent experiments. 

Similar results were achieved by using HT (Figure 2B). The mean lifespan after heat stress was increased by about 11% by treatment with 100 µg/mL HT, by 22% with 250 µg/mL HT and by 14% with 500 µg/mL HT. There was no significant difference between the survival curves of these treatment groups among each other. Again, 30 µg/mL was not sufficient to improve the survival. Based on these results, 250 µg/mL HT was applied for further experiments. None of the tested concentrations and compounds exerted a harmful effect on the survival of the nematodes after stress exposure. 

### 2.2. Hydroxytyrosol Prolonged the Lifespan of Wild Type *C. elegans*

Several olive polyphenols and preparations have proven to be effective in extending the lifespan in *C. elegans* [57,58]. However, the olive oil polyphenols investigated in this study have not been tested so far in this sense. 

Surprisingly, OLE treatment did not result in any significant lifespan enhancement (Figure 3A): The mean lifespan of wild type nematodes was only hardly noticeably increased by 2.7%, which is probably the result of a minor, not significant, increase in the median lifespan from 22.55 days to 23.31 days (Table 1). However, the treatment with HT led to an increase of mean lifespan by 14.1% (Figure 3B). This life prolongation was not only visible in terms of mean and median lifespan, but was also reflected in terms of minimum and maximum lifespan (the time point, when 25% or 90%, respectively, of the individuals are dead) as well as in the time point when final death occurred (Table 1). No obvious side effects, such as extrusion of internal organs through the vulva or morphological alterations in movement in the polyphenols-treated groups compared to the controls were seen during lifelong observation.

### 2.3. Polyphenols Improved Age Pigment Accumulation and Locomotive Behaviour in Wild Type Nematodes

The intestinal autofluorescence, one of the most prominent ageing and health biomarker in *C. elegans* [49], was surveyed to determine the overall health status. Observations with a fluorescence microscope fitted with a red filter set allowed to detect the accumulation of the “age pigment” in different age classes. As expected, the red fluorescent intensity increased with age (Figure 4), whereas the increase was only weakly pronounced at the 7^th^ day of adulthood. Both polyphenolic compounds were able to reduce age-related gain in autofluorescence. The quantity of fluorescent pigments was slightly, yet significantly, diminished at the 12^th^ day, but not at the 3^rd^ or 7^th^ day, of adulthood (Figure 4). 

Since ageing is marked by physical decline, sarcopenia is considered a valuable parameter of health status in organisms of metazoans, including *C. elegans* [59,60]. Therefore, the ability of both polyphenols to boost the health of nematodes was additionally assessed with a swim assay. We measured the thrashing rate, the body wave number and the activity index to determine the physical performance at different ages. The thrashing rate is the number of body thrashes per minute as an indicator for the speed of movement whereas the activity index sums up the number of pixels that are covered by the body during the time needed for two strokes as an indicator for the vigorousness of bending [61]. Furthermore, the body wave number, a feature that increases with age, determines the waviness of the body at each time point. Indeed, the vigorousness and speed of movement of untreated worms declined with age, as indicated by the differences in both endpoints between the 3^rd^ and 7^th^ or 12^th^ day of adulthood, respectively (Figure 5A,C). Moreover, the body wave number increased with age (Figure 5B), in agreement with the results from Restif et al. [61], thus verifying the correct performance of the test. 

OLE treatment resulted in a remarkable increase of the number of thrashes per minute (Figure 5A) and of the activity index (Figure 5C) displayed by worms at all three tested stages, whereas the body wave number was decreased at the 7^th^ and 12^th^ day of adulthood (Figure 5B). The percentage increase was about 23% (A3), 89% (A7) and 64% (A12) and about 49% (A3), 69% (A7) and 61% (A12) for thrashing rate and activity index, respectively. The decrease of the body wave number reached its maximum at the 7^th^ day of adulthood with a reduction of 39%. Surprisingly, HT was not able to enhance the thrashing rate of the nematodes at any adult-day (Figure 5A). However, at the 12^th^ day of adulthood, an increase of 43% was detected by analysing the covered pixel per body and minute (Figure 5C). In addition, a decrease of 25% in the body wave number was found at A12 as well (Figure 5B). No polyphenol led to a reduction of motor performance in treated worms.

### 2.4. *C. elegans* Parkinsonian Models Profit from Olive Oil Polyphenol Treatments

Exposure to the pesticide rotenone or the transgenic expression of human α-synuclein induces the Parkinsonian-like syndrome in C. *elegans*, which manifests in impaired movement [62,63]. In order to assess whether the polyphenols are able to reduce this symptom, the swimming analysis was performed with nematodes suffering from Parkinson’s-like symptoms induced by exposure to rotenone of wild type worms. Furthermore, transgenic Parkinson’s models expressing human α-synuclein in muscle cells (strain OW13) and dopaminergic neurons (strain UA44), respectively, were treated with polyphenols as well.

Treatment with 10 µM rotenone led to dramatically decreased movement abilities. Given that untreated nematodes exhibit more than 50 thrashes per minute at A3, the thrashing rate was reduced by more than 80% to less than 10 thrashes per minute in rotenone-exposed C. *elegans* (Figure 6A). Similar proportions could be monitored for the activity index of young control nematodes with or without rotenone treatment (Figure 6C). Furthermore, the body wave number was more than doubled due to exposure to rotenone compared to untreated nematodes (Figure 6B). Interestingly and in contrast to rotenone-untreated nematodes, the locomotion abilities after exposure to rotenone did not decline after the 3^rd^ day of adulthood (Figure 6). OLE was able to partly inhibit the rotenone-induced movement decline in both tested ages and all swim traits by more than doubling the measured values (Figure 6A,C) or by reducing them by at least 42% (Figure 6B). Quite strong effects were also visible by using HT in all tested ages and endpoints: HT increased the thrashing rate by at least 56% and the activity index by a minimum of 116%. The body wave number was decreased by at least 23%. 

To check the anti-Parkinson’s effect of olive oil polyphenols, the motor activity was also assessed in the transgenic C. *elegans* OW13 and UA44 strains, both models of synucleinopathies. The presence of α-synuclein in the body wall muscle cells (strain OW13) leads to movement deficits, as previously described by Van Ham, et al. [64]. Indeed, the thrashing rate clearly deteriorated from about 50 thrashes/min in young untreated wild type adults to about 35 and 23 thrashes per minute, respectively, in young untreated OW13 worms (Figure 7A). This effect, albeit weaker, was also seen, in part, in older nematodes as well as for the activity index (Figure 7C). The body wave number was very stable between the 3^rd^ and 7^th^ day of adulthood (Figure 7B). Interestingly, control DMSO treatment already led to a mild beneficial effect for all three swim parameters in the OW13 strain. 

Treatment with either polyphenol improved several swim performance features in this strain over control treatments (Figure 7A–C). OLE administration provoked a 27% increase of thrashes per minute at day 3 and a 40% increase at day 7 (Figure 7A), whereas the increase of the activity index (27%) was detected only at day 7 (Figure 7C) and no significant change was seen in the body wave number (Figure 7B). HT displayed its advantageous effects in both tested age groups and in all swim parameters (Figure 7A–C), whereas HT remarkably increased the thrashing rate by 71% in A3. 

Interestingly, the strain UA44, which is characterized by the presence of α-synuclein in dopaminergic neurons, did only show a negligible decline in locomotion from day 3 to day 7 of adulthood (Figure 8). Furthermore, the difference in swim performance between young wild type and young UA44 nematodes was hardly recognizable, a finding that agrees with data from a previous study, showing that α-synuclein controlled by a dopaminergic promotor did not disturb the thrashing speed [65]. Surprisingly, treatment with either polyphenol resulted only in a limited improvement in swim behaviour of the UA44 strain. OLE treatment increased the activity index in young nematodes (Figure 8C) but resulted only in minor and non-significant changes of the thrashing rate and body wave number for UA44 (Figure 8A,B). On the other hand, HT supplementation showed beneficial effects only on the number of thrashes per minute in older worms (Figure 8A), but not on the magnitude of movement or the waviness (Figure 8B,C).

To summarize, HT and OLE treatments resulted in enhanced swim performance in nematodes suffering from rotenone exposure or α-synuclein expression in muscle cells (strain OW13). However, the strain UA44, characterized by α-synuclein expression in dopaminergic neurons, only weakly profited from the polyphenol treatments. 

### 2.5. α-synuclein Induced Damages and α-synuclein Accumulation was Targeted by Olive Oil Polyphenols In Vivo

The advantage of the C. *elegans* OW13 strain is the yellow fluorescent labelling of synthesized α-synuclein driven by the muscle specific *unc-54*-promoter. Therefore, the potential polyphenolic inhibition of the pathological α-synuclein accumulation in muscles can be observed via fluorescence microscopy. Treatment of this model with either polyphenol resulted in a progressive reduction of α-synuclein accumulation in the body wall muscle cells compared to the control groups: The reduction of α-synuclein accumulation was about 5% at day 3 and 8% at day 7 and 12 of adulthood in OLE-treated groups (Figure 9). Even more pronounced effects were monitored by using HT, with a reduction of α-synuclein accumulation by 6% at day 3, 7% at day 7, and 14% at day 12 of adulthood. Overall, the fluorescence intensity declined with age, a finding in line with the ageing-dependent decline of *unc-54* expression observed by Adamla and Ignatova [66].

In addition to the OW13 strain, the UA44 transgenic animals were also monitored under fluorescent conditions. The α-synuclein accumulation in dopaminergic neurons causes neurodegeneration in the UA44 transgenic strain [63]. Moreover, GFP linked to the dopamine transporter in dopaminergic nerve cells allows to visualize the quantity and quality of the six anterior and two posterior dopaminergic neurons after α-synuclein-induced damage [67]. 

Counting and description of each dopaminergic neuron was performed as proposed by Harrington, et al. [68]. We counted six anterior dopaminergic neurons (four CEP neurons and two ADE neurons) and two posterior DA neurons (PDE neurons) [69]. Every type of alteration, such as the loss of uniformity of the neuronal body and the reduction of the fluorescence up to neuronal shutdown, were noted in order to classify the neurons as degenerated (as shown in Figure 10B) or intact (Figure 10C). The fraction of worms with damaged dopaminergic neurons was growing with age (Figure 10A), however, HT was able to minimize neuronal damages especially in older nematodes (Figure 10A). A smaller and non-significant neuroprotective effect on dopaminergic neurons was also obtained with OLE treatment (Figure 10A). 

## 3. Discussion

### 3.1. HT and OLE Boost the Health, but not Necessarily the Lifespan, of Wild Type C. elegans

Lifespan analyses in *C. elegans* treated with plant polyphenols and other natural compounds were often reported with positive outcome (reviewed in Collins, et al. [70] and Pallauf, et al. [71]). The mechanisms behind this beneficial action are discussed in different directions, ranging from antioxidant, pro-oxidant, hormetic, direct pathway targeting, or caloric restriction mimetic effects, to recall only a few [72,73,74,75,76]. In the present study, HT was shown to improve the lifespan of *C. elegans* as well as the accumulation of age-pigment and swim behaviour in old worms. Thus, health and lifespan were targeted in parallel as expected. Interestingly, OLE also exerted very good performances regarding the impact on locomotion, stress resistance and age-pigment accumulation, yet in the absence of a life extending effect. However, since longer life does not automatically indicate healthier life [77,78], the suggestion that vice versa improved health is not a guarantee for a longer life, is not unreasonable. 

Healthspan is hard to define and a lot of parameters need to be considered, but, simplifying, it can be described as the period of life in which the individual is functionally independent and free from serious diseases [79,80]. Uncoupling of the correlation between lifespan and healthspan was discussed in detail for *daf-2 C. elegans* knockout mutants, in which the lifespan-extending inhibition of insulin signalling did [60,81] or did not [78] increase healthspan in parallel. However, this discussion is not restricted to worms, but also takes place for flies, mice and humans [77,82,83]. Thus, it remains unclear how and to what extent healthspan and lifespan correlate. Furthermore, the different outcomes in the labs as exemplified by *daf-2* underline that healthspan-related features should be tested in a standardized way; not only to maintain comparability between results from different labs, but also to fully characterize the potential relationship between lifespan and healthspan in an objective manner. 

Nevertheless, here the question arises, why OLE only affected healthspan but not lifespan? The stability of OLE in aqueous and ethanol solutions was shown to be better than other polyphenols [84,85] and during UV-induced degradation, the life-extending hydroxytyrosol is one of the main end products of its metabolism. Thus, an elevated level of OLE degradation during long-lasting lifespan analysis is not a sufficient explanation, also because such an effect should be much stronger for other, less stable polyphenols like quercetin (reviewed in Wang, et al. [86]), which it is not [87,88]. Another explanation is provided by the mode of action of the green tea ingredient epigallocatechin 3-gallate (EGCG). Brown et al. [89] and Zhang et al. [90] reported that, in spite of several health benefits in EGCG-treated worms, including prolonged survival under stress, no survival advantages were monitored during standard culture conditions. The antioxidant capacity of EGCG was emphasized as the main biochemical mechanisms responsible for the improvement of diverse health attributes. Due to the known strong antioxidant characteristics of OLE [91], this could also be true for OLE; accordingly, the missing lifespan prolongation by EGCG and OLE is not unexpected considering that Pun et al. [92] observed that antioxidant actions do not lead to longevity in *C. elegans*. Based on this consideration, the action of HT needs to be reconsidered. It must be concluded that the lifespan extension by HT is probably independent of its antioxidant power. 

Finally, it needs to be mentioned that all tests were performed in the presence of 5-fluoro-2-deoxyuridine (FUdR). Since FUdR was shown to have (mainly positive) influences on the stress resistance and lifespan in *C. elegans* [93,94,95], it cannot be excluded that this could lead to false-negative results. This could be an alternative explanation for the missing life-prolonging effects in the OLE-treated group. 

### 3.2. Anti-Parkinson’s Syndrome Effects: Evidence from Three Different PD Models of *C. elegans*

Cooper, et al. [96] described that decelerating ageing may provide a possible treatment for PD. The beneficial effects of OLE and HT on the age-related intestinal autofluorescence and on the locomotion in old worms are strong indicators that the ageing process itself is reduced by these polyphenols. Thus, the anti-PD action of OLE and HT was not entirely surprising. Indeed, both polyphenols showed strong and convincing anti-PD activity. However, which is the underlying mechanism?

DA neurons are suffering from oxidative stress in the rotenone PD model because this pesticide generates reactive oxygen species [62]. Therefore, the neuroprotective role of the analysed polyphenols in the *C. elegans* rotenone model, characterized by defects in swimming behaviour, may be related to their antioxidant activity. Indeed, the treatment and prevention of PD with antioxidants was discussed and tested repeatedly [14,97,98], but the results were sobering, suggesting that antioxidant treatments might not be the key to combat PD [99]. 

The protective properties of plant polyphenols on DA neurons could also be associated not only with the structure of HT, also a product of dopamine metabolism, but also to its ability to induce phase II detoxifying enzymes. These include NADPH quinone oxidoreductase-1, heme oxygenase-1, glutathione S-transferase, and the modifier subunit of glutamate cysteine ligase which catalyses the first step in the synthesis of GSH [22]. Indeed, a *Drosophila* PD model was used to show that boosted phase II enzyme activity reduces α-synuclein-mediated neuronal loss [100]. 

Interestingly, both polyphenols reduced, age-dependently, the build-up of human α-synuclein in the body wall muscle cells of a transgenic *C. elegans* model (strain OW13) and improved swim performance. However, it is not clear whether the polyphenols are able to eliminate accumulated α-synuclein or prevent its accumulation and which mechanistic process is responsible for their action. One further idea regarding their mode of action is delivered by Angeloni et al. [22] who showed that polyphenols are able to modulate the proteostatic machinery, both at the proteasome complex and at the autophagic flux. More specifically, one study evaluated the effect of OLE on the proteasome complex of human embryonic fibroblasts, showing an improvement of the proteasome activity [101]. Since the impairment of the ubiquitin-proteasome complex is deeply involved in the pathogenesis of neurodegenerative diseases [102], the induction of proteasome activity might be a possible reason for the anti-PD effects of polyphenols. Other studies with OLE and HT reported a remarkable activation of the autophagic flux in a murine model of Aβ deposition, with a significant reduction of plaque load following activation of microglia [38,39]. These data also support the anti-PD activity of these polyphenols following reduction of α-synuclein aggregates in the affected brain areas.

To validate the neuroprotective action of polyphenols on dopaminergic neurons, experiments were performed also in another *C. elegans* model of PD, the strain UA44, where α-synuclein induces qualitative and quantitative damages of the six anterior and two posterior dopaminergic neurons. Our data showed that HT was able to minimize neuronal damage especially in older nematodes. 

As reported above, HT is also endogenous to the brain as a catabolite of neurotransmitter breakdown. The neurotoxic action by dopamine and its intermediate metabolites is described as an autotoxic mechanism that contributes to the selective loss of dopaminergic neurons in PD [3]. HT, also known as DOPET (3,4-dihydroxyphenylethanol), is produced from dopamine by dopamine oxidative metabolism [103] in order to reduce the levels of the neurotoxic intermediate product 3,4-dihydroxyphenyl-acetaldehyde (DOPAL) in dopaminergic neurons [104]. Whether this metabolic pathway is connected to the observed beneficial action of exogenous HT remains an open question. A smaller and non-significant neuroprotective effect on dopaminergic neurons also resulted from OLE treatment. In future studies, the question needs to be answered, whether the longevity effect and the beneficial effects in the UA44 strain following HT treatment are based on the same mechanisms, which are not present or weaker in OLE-treated worms. 

Tyrosol differs from HT only by one hydroxyl group; it was also shown to be a potent health- and lifespan-boosting substance in *C. elegans* [58]. However, in contrast to HT, tyrosol could not exert any preventive effects in kidney cells subjected to oxidative stress [105]. The beneficial action of HT compared to that of tyrosol could be explained by the higher scavenging and antioxidant activity of HT due to the additional hydroxyl group [106]. Nevertheless, this increased antioxidant power seems not to be the only mode of action as suggested by the convincing antioxidant-study in *C. elegans* from Pun et al. [92], by additional studies, questioning the power of antioxidants, reviewed in [107] as well as by the mitohormesis concept, which underlines the importance of ROS [108,109]. 

Another possible background mechanism for the observed effects could be hormesis, already considered in the case of the effect of tyrosol in *C. elegans* [58]. Several studies indicate that different stressors extend lifespan in *C. elegans* in a hormetic-like manner [110,111,112] and suggest that hormetic effects could be exploited to prevent the onset of various diseases [13,113,114], including neurodegenerative disorders, and to slow down the ageing process [115,116]. 

In addition, a recent study demonstrated that hormetic dietary phytochemicals might improve health and extend lifespan through mild elevation of ROS, which activate a number of stress adaptive genes in *C. elegans* via HSF-1 and SKN-1/Nrf2 signalling pathways [117]. Govindan and colleagues also note that the hormetic stress by phytochemicals suppresses the late age onset of misfolding and aggregation of proteins such as α-synuclein in PD. The close link between stress and ageing suggests that interventions harnessing the hormetic mechanisms may extend lifespan or delay age-associated functional decline. Taken together, these data indicate that low concentrations of natural polyphenols such as OLE and HT generate a moderate functional stress that extends healthspan in wild type and experimental models of PD. This is consistent with the idea that “neurohormesis” may have anti-ageing effects thanks to the induction of adaptive pathways triggered to cope with a mild neuronal stress, opening novel potential therapeutic strategies for clinical interventions against the onset and/or progression of PD in humans.

## 4. Materials and Methods

### 4.1. C. elegans Strains and Culture Conditions

The wild type *C. elegans* strain N2 (Var. Bristol), the transgenic *C. elegans* strain OW13 (*grk-1*(ok1239), pkIs2386 [*unc-54*p::α-synuclein::YFP + *unc-119*(+)]), as well as the *E. coli* OP50 feeding strain were obtained from the Caenorhabditis Genetics Centre, University of Minnesota (Minneapolis, MN, USA). The *C. elegans* strain UA44 (baIn11[P*dat-1*::α-synuclein, P*dat-1*::GFP]) was kindly provided by the Caldwell laboratory, University of Alabama (Tuscaloosa, AL, USA) [118]. *C. elegans* wild type and transgenic strains were grown on standard nematode growth medium (NGM) at 22 °C, seeded with *E. coli* OP50 and maintained following standard protocols as described previously [119]. Prior to all tests, synchronous L1 larvae were obtained via “egg prep” by strongly shaking at least 2000 young adults for about 4 min in 10 mL bleaching solution (0.5 mL NaOH (10 M), 2.5 mL sodium hypochlorite-solution (12%), and 7 mL bidest water). After washing with M9 buffer at least three times, the resulting egg pellet was slightly shaken overnight in 4 mL M9 buffer. The following day, the hatched L1 larvae were distributed to NGM plates seeded with OP50. Forty-eight hours later, L4 larvae were transferred to treatment plates as described below. 5-fluoro-2-deoxyuridine (FUdR; Tokyo Chemical Industry, Eschborn, Germany) was used to inhibit fertilization [120] and was dropped onto each plate with a final concentration of 100 µM (according to agar volume). 

### 4.2. Polyphenol and Rotenone Treatment

The treatment plates were prepared with the polyphenols oleuropein aglycone (Extrasynthese) and hydroxytyrosol (Sigma-Aldrich, St. Louis, MO, USA). Glycated oleuropein was de-glycosylated by treatment with almond β-glucosidase (EC 3.2.1.21, Fluka, Sigma Aldrich, St. Louis, MO, USA) as previously described [121], with minor modifications. Briefly, 10 mM oleuropein in 0.1 M phosphate buffer (pH 7.0) was incubated overnight with β-glucosidase (8.9 I.U.) at RT. Then, the reaction mixture was centrifuged, the precipitate re-suspended in 50% (*v*/*v*) dimethyl sulfoxide (DMSO) and the solution kept frozen. Complete deglycosylation was assessed by assaying the released glucose. HT powder was dissolved in bidest water at 60 mg/mL and the solution stored at −20 °C. OLE and HT were added to the bacteria and agar at a final concentration of 30 μg/mL, 100 μg/mL, 250 μg/mL, or 500 μg/mL, respectively. A final concentration of 0.05% DMSO (for OLE assays) or equal amounts of bidest water (for HT assays) were applied in all treatment and control plates as well as feeding bacteria. 

To trigger the Parkinsonian-related phenotype, wild type nematodes were treated with rotenone (Sigma-Aldrich, St. Louis, MO, USA). A stock solution of 0.5 mg/mL rotenone was prepared in DMSO and added with a final concentration of 10 µM to the control and polyphenol plates. After distribution with a spatula and drying for 24 h in the dark, OP50, including 10 µM rotenone and the respective polyphenol, was spread on the plates. L4 nematodes were transferred to the rotenone plates until they were used for bioassays.

### 4.3. Lifespan and Heat Stress Assay

Synchronized wild-type nematodes were observed and scored for their survival throughout their lifespan. The animals were counted daily from the first day of adulthood until all died. To assess the viability of the nematodes, they were first gently touched with a platinum wire at the tail and the head. If no movements were recognizable, the pharyngeal pumping was observed. The worms were considered dead when none of these movements were recognizable. The nematodes at the edge of the plate or in the depth of the agar were considered lost and excluded from the count.

The heat stress test was performed according to the lifespan protocol with the difference that nematodes at the third day of adulthood were stressed for 3 h at 37 °C and counting of dead and alive worms started 1 day after stress exposure. 

### 4.4. Fluorescence Microscopy Analysis

For the fluorescence observation, the nematodes were placed on a 2% agar pad on a microscope slide and anesthetized with 4 µL NaN_3_ (1M). The images were taken with the aid of the Axioskop fluorescence microscope (Carl Zeiss, Oberkochen, Germany) and filter set 13 from the Zeiss 4880 series (Carl Zeiss, Oberkochen, Germany). Nematodes with a ruptured vulva phenotype were excluded from analysis. The images were captured and analysed on the 3^rd^, 7^th^ or 12^th^ day of adulthood, respectively.

Wild type nematodes were analysed to determine and quantify ageing-related pigment accumulation. The images were captured at 100x magnification and a red filterset (TRITC, 545/30 nm ex, 610/70 nm em) was used. It was necessary to acquire additional images in a bright field, because of the poor visibility of the body contour in the dark field. The images in the bright field were used to delineate the perimeter of the worm to which the dark field image was overlayed by the CellProfiler Software (Version 3.1.9; Broad Institute, Cambridge, MA, USA) [122,123]. The quantification of age-related pigment accumulation was expressed by the mean intensity of the red fluorescence per total worm body.

The OW13 transgenic strain features yellow fluorescent protein linked to α-synuclein in the body wall muscle cells. Therefore, the nematodes were monitored using a yellow barrier filter with 100x magnification to quantify the light emission proportional to the amount of accumulation of the pathological protein. The images were processed using the CellProfiler software, and the yellow fluorescence intensity emitted per total worm body was calculated.

The UA44 transgenic strain features GFP linked to the dopamine transporter in the six dopaminergic neurons of the head and two in the tail as well as harmful α-synuclein in dopaminergic neurons. The green fluorescence intensity represents the vitality of the neurons, therefore, the green barrier filter was used for the analysis. The number of detectable anterior neurons was counted at the microscope with 200x magnification. In addition, individual images of the head were captured. The nematodes were assayed for patterns of degeneration at indicated time points, as described previously from Harrington, et al. [68].

### 4.5. Swim Behaviour Assay

The study of locomotion was realized with a swim assay according to Restif et al. [61] and Ibáñez–Ventoso et al. [124]. Wells with a depth of 0.5 mm and Ø 10 mm were created with two self-adhesive marking films for microscope slides and filled with M9 buffer. Five to 10 nematodes were transferred per well and covered by a cover slip to facilitate visualization and recorded with a connected camera. Nematodes with a ruptured vulva phenotype were excluded from analysis. The analysis was conducted with wild type nematodes at the 3^rd^, 7^th^, and 12^th^ day of adulthood. Furthermore, wild type rotenone-treated worms and the strains OW13 and UA44 were analysed at the 3^rd^ and 7^th^ day of adulthood. Each video was converted into single frames and processed with Photoshop (version 19.1.7; Adobe Inc., San José, CA, USA) to meet the required settings. Thereafter, pictures were analysed with the CeleST software (version 3.1; distributed by GitHub Pages, https://github.com/DCS-LCSR/CeleST, accessed on 08 April 2020) as described by Restif et al. [61] and Ibáñez–Ventoso, et al. [124]. The thrashing rate, the body wave number and the activity index were evaluated as representative parameters of motor activity.

### 4.6. Statistical Analysis

All experiments were independently conducted at least two times. The Online Application for Survival Analysis (OASIS 2; https://sbi.postech.ac.kr/oasis2/, accessed on 8 April 2020) [125] was used for comparing survival differences between two conditions. Fluorescence intensities as well as swim behaviour parameters were calculated as mean ± SEM, and statistical significance was calculated by a two-tailed t-test using GraphPad (https://www.graphpad.com/quickcalcs/, accessed on 8 April 2020). Chi-square test was used to compare the number of worms with intact and degenerated neurons in the UA44 strain. 

## 5. Conclusions

Due to their different actions in terms of lifespan during non-stressful conditions, disparate modes of actions could be the underlying cause of the beneficial characteristics of the olive polyphenols OLE and HT. Accordingly, possible additive or even synergistic effects by combining both polyphenols should be studied in the future. Intense research in the last decade has provided increasing knowledge of the biochemical and cell biology basis of the beneficial effects of plant, notably olive polyphenols. Such growing information indicates that these polyphenols have the potential as promising tools to be used to develop new therapeutic and preventive approaches against ageing and ageing-associated neurodegenerative diseases. However, even though *C. elegans* and mammalian models are frequently used to test possible human treatments, human clinical trials are still needed to verify this assumption.

## Figures and Tables

**Figure 1 ijms-21-02588-f001:**
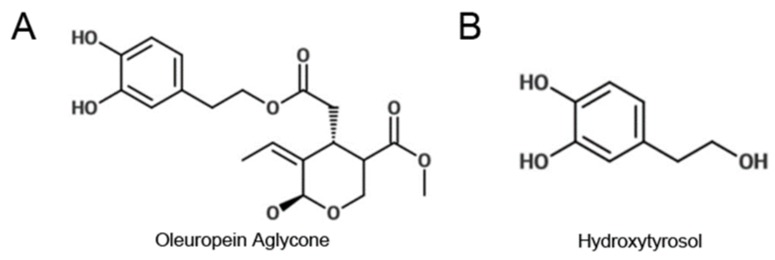
Chemical structure of (**A**) oleuropein aglycone and (**B**) hydroxytyrosol.

**Figure 2 ijms-21-02588-f002:**
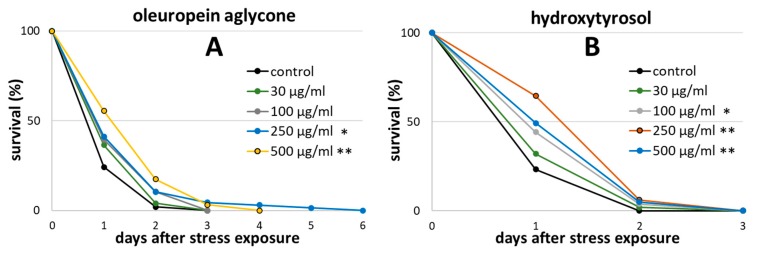
Heat stress survival in presence of different OLE and HT concentrations. Survival is expressed as a percentage of the initial population combined with three biological replications. At the third day of adulthood (day 0), wild type nematodes were exposed to heat stress at 37 °C for 3 h prior monitoring survival. (**A**) Survival curves during OLE treatment with *n* (control) = 70, *n* (30 µg/mL) = 74, *n* (100 µg/mL) = 44, *n* (250 µg/mL) = 68, and *n* (500 µg/mL) = 64; (**B**) Survival curves during HT treatment with *n* (control) = 90, *n* (30 µg/mL) = 50, *n* (100 µg/mL) = 52, *n* (250 µg/mL) = 67, and *n* (500 µg/mL) = 79. Statistical significance was calculated by log-rank test including Bonferroni correction. Differences compared to control were considered significant at *p* < 0.05 (*) and *p* < 0.001 (**). *n*: number of tested nematodes.

**Figure 3 ijms-21-02588-f003:**
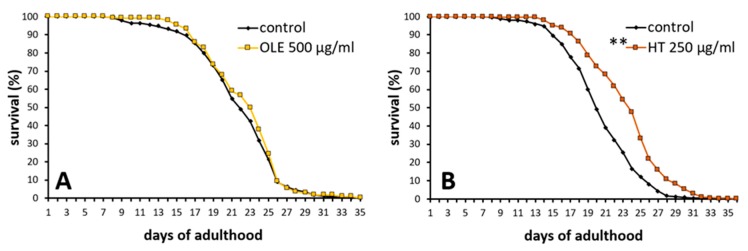
Effect of OLE and HT on lifespan in *C. elegans*. The survival curves of controls and polyphenol-treated nematodes are shown. Survival is expressed as a percentage of the initial population per day. (**A**) The curve represents two independent experiments (*n* control = 184, *n* OLE = 111); (**B**) Representative survival curve of three independent experiments with control and HT-treated worms (*n* control = 250, *n* HT = 286). Statistical significance was calculated by log-rank test; differences compared to control were considered significant at *p* < 0.05 (*) and *p* < 0.001 (**). *n*: number of tested nematodes.

**Figure 4 ijms-21-02588-f004:**
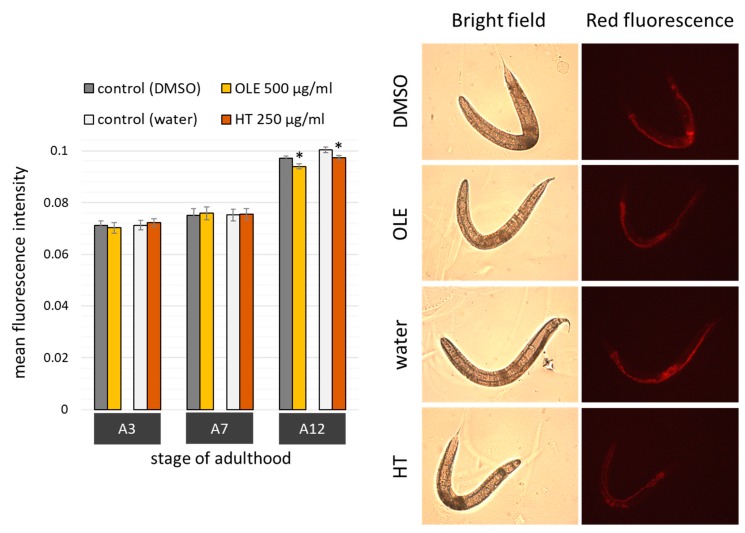
Age pigment quantification after OLE and HT treatment in *C. elegans*. Nematodes were observed by fluorescence microscopy in red spectrum at day 3, 7, and 12 of adulthood in two biological repeats. The bar charts (left) show the mean red fluorescence intensity of OLE (*n* A3 = 34; *n* A7 = 38; *n* A12 = 88) and DMSO (*n* A3 = 39; *n* A7 = 43; *n* A12 =84)-treated nematodes as well as HT (*n* A3 = 38; *n* A7 = 45; *n* A12 = 94)-treated nematodes and their respective water control (*n* A3 = 38; *n* A7 =44; *n* A12 = 95). Data are represented as mean ± SEM, and statistical differences compared to control were considered significant at *p* < 0.05 (*). *n*: number of tested nematodes; A3, A7, A12: 3^rd^, 7^th^ and 12^th^ day of adulthood. In addition, example pictures (right) representing bright field and red fluorescence shots at the 12^th^ day of adulthood in the control (DMSO and water) and polyphenol-treated (OLE and HT) groups are shown (all scale bars = 200 µm).

**Figure 5 ijms-21-02588-f005:**
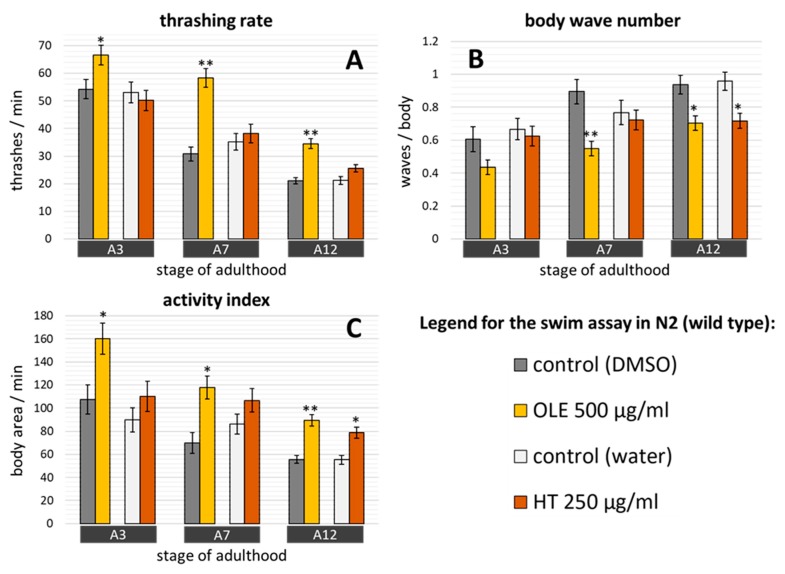
Swim behaviour characteristics in wild type *C. elegans* treated with OLE and HT. Locomotory performances were determined at day 3, 7, and 12 of adulthood in two independent repeats. The determination of locomotion differences comprises three parameters: (**A**) the thrashing rate, (**B**) the body wave number, and (**C**) the activity index. The number of analysed animals accounts for: DMSO control = 67 (A3), 70 (A7) and 117 (A12); OLE = 63 (A3), 70 (A7) and 106 (A12); water control = 76 (A3), 70 (A7) and 111 (A12); HT = 70 (A3), 70 (A7) and 104 (A12). Data are presented as mean ± SEM and differences compared to control were considered significant at *p* < 0.05 (*) and *p* < 0.001 (**). A3, A7, A12: day 3, 7, 12 of adulthood.

**Figure 6 ijms-21-02588-f006:**
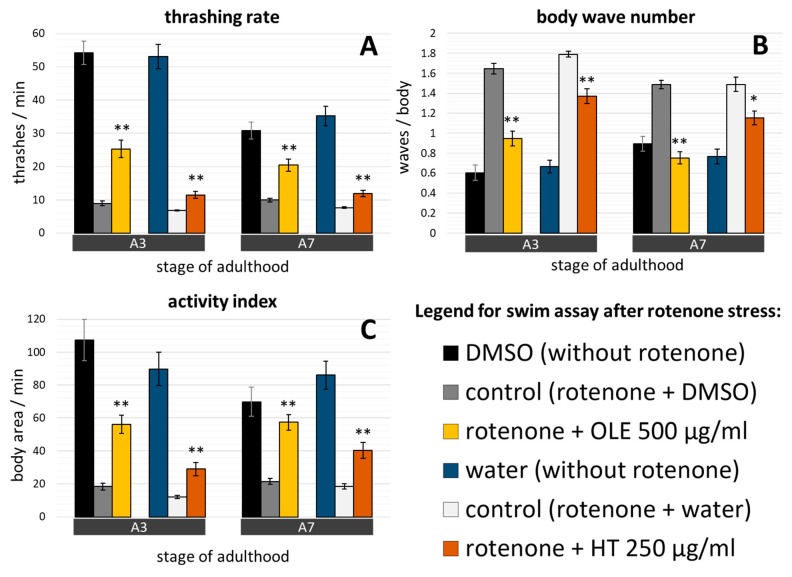
Effect of OLE and HT on rotenone-induced locomotion deficits. The administration of 10 µM rotenone, starting at the fourth larval stage, led to Parkinsonian-like phenotype exhibiting movement impairments at the 3^rd^ and 7^th^ day of adulthood. (**A**) The thrashing rate, (**B**) the body wave number, and (**C**) the activity index are shown with and without simultaneous OLE or HT administration. The number of tested nematodes was: DMSO control = 58 (A3) and 69 (A7); OLE = 60 (A3) and 63 (A7); water control = 62 (A3) and 46 (A7); HT = 44 (A3) and 61 (A7). A3, A7: day 3 and 7 of adulthood. Data are pooled from two biological repeats and presented as mean ± SEM, and differences compared to control were considered significant at *p* < 0.05 (*) and *p* < 0.001 (**). To enable direct comparisons, data from nematodes without rotenone and polyphenol exposures (see Figure 5) are shown in addition.

**Figure 7 ijms-21-02588-f007:**
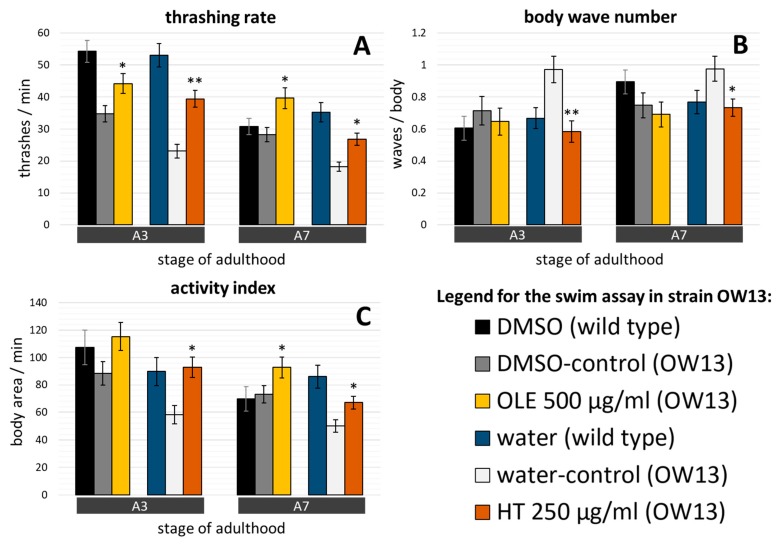
OLE and HT effects on the swim performance in the OW13 strain. (**A**) The thrashing rate, (**B**) the body wave number, and (**C**) the activity index were determined at day 3 and day 7 of adulthood with and without OLE or HT treatment. Here, the nematode strain OW13, characterized by α-synuclein in the body wall muscle cells, was used. The number of tested nematodes was: DMSO control = 65 (A3) and 71 (A7); OLE = 61 (A3) and 60 (A7); water control = 51 (A3) and 71 (A7); HT = 61 (A3) and 66 (A7). A3, A7: day 3 and 7 of adulthood. Data are collected in two independent trials and are presented as mean ± SEM. Differences compared to control were considered significant at *p* < 0.05 (*) and *p* < 0.001 (**). To enable direct comparisons, data from wild type nematodes without polyphenol exposures (see Figure 5) are shown in addition.

**Figure 8 ijms-21-02588-f008:**
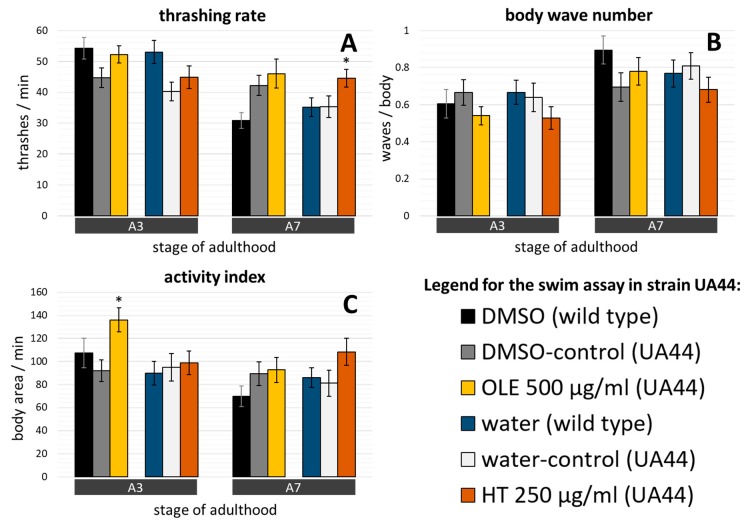
Effect of OLE and HT on the swim performance in the *C. elegans* UA44 strain. (**A**) The number of thrashes per minute, (**B**) the number of waves running through the body, and (**C**) the area covered by the body per minute were observed in presence and absence of OLE and HT at day 3 and 7 of adulthood in two biological repeats. The UA44 strain used in this study is characterized by α-synuclein in dopaminergic neurons. The bar charts represent the following number of individuals: DMSO control (*n* A3 = 63, *n* A7 = 50), OLE (*n* A3 = 72; *n* A7 = 55), water control (*n* A3 = 57, *n* A7 = 55) and HT (*n* A3 = 48, *n* A7 = 65). *n*: number of tested nematodes. A3, A7: day 3, 7 of adulthood. Data are represented as mean ± SEM and differences compared to control were considered significant at *p* < 0.05 (*). To enable direct comparisons, data from wild type nematodes without polyphenol exposures (see Figure 5) are shown in addition.

**Figure 9 ijms-21-02588-f009:**
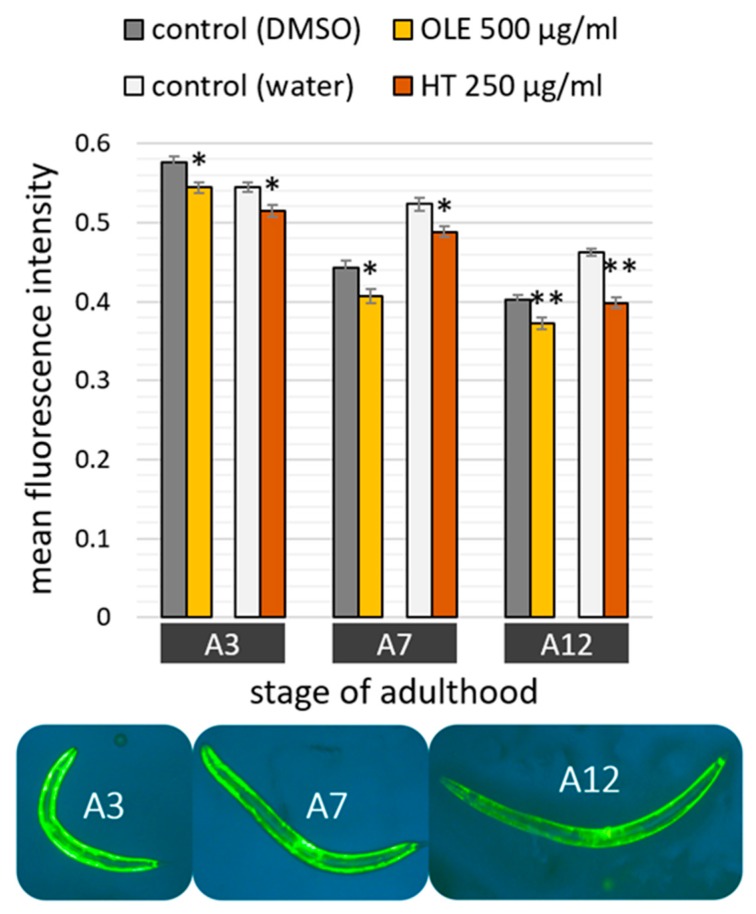
Changed α-synuclein accumulation in muscle cells of the OW13 strain after OLE and HT treatment. Nematodes from the OW13 strain were subjected to polyphenolic treatments starting from L4 and analysed by fluorescence microscopy using a yellow filter at day 3, 7, and 12 of adulthood. The bars represent the mean fluorescent intensity ± SEM from two biological repeats and the number of tested nematodes were: DMSO control (*n* A3 = 32, *n* A7 = 48, *n* A12 = 38), OLE (*n* A3 = 28; *n* A7 = 43, *n* A12 = 30), water control (*n* A3 = 47, *n* A7 = 75, *n* A12 = 66) and HT (*n* A3 = 35, *n* A7 = 39, *n* A12 = 44). A3, A7, A12: day 3, 7, 12 of adulthood. Differences compared to control were considered significant at *p* < 0.05 (*) and *p* < 0.001 (**). In addition, three example pictures from untreated OW13 nematodes visualising the age-dependent fluorescent change are shown (all scale bars = 200 µm).

**Figure 10 ijms-21-02588-f010:**
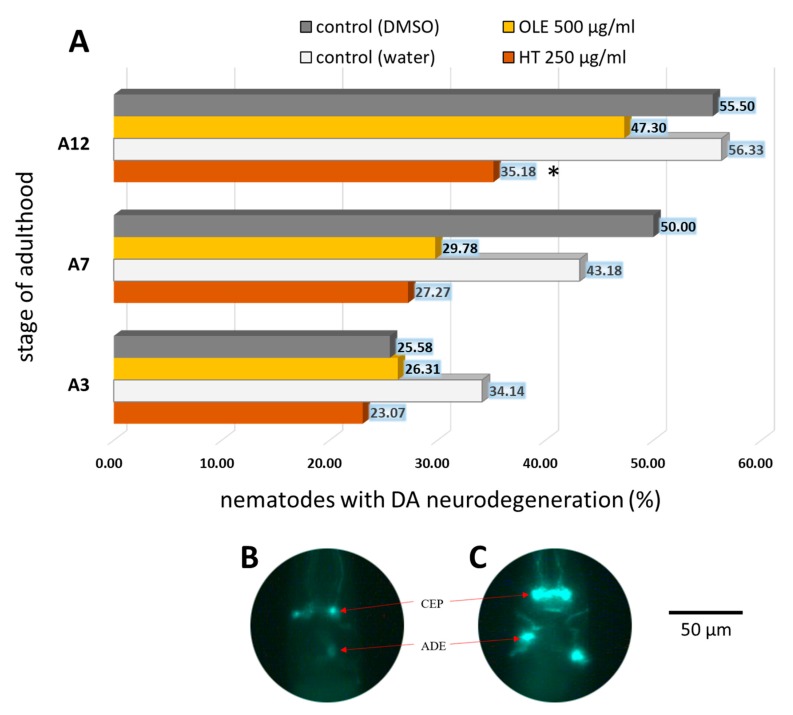
The impact of OLE and HT on dopaminergic neurodegeneration caused by α-synuclein in the *C. elegans* UA44 strain. (**A**) Shown are the percentages of nematodes with degenerated dopaminergic anterior neurons with and without polyphenolic treatment. The determination of the type and frequency of aberrations of dopaminergic neuronal viability was performed at day 3, 7, and 12 of adulthood. The number of tested nematodes in three biological repeats was: DMSO control = 43 (A3), 31 (A7) and 45 (A12); OLE = 38 (A3), 47 (A7) and 39 (A12); water control = 41 (A3), 44 (A7) and 71 (A12); HT = 39 (A3), 44 (A7) and 54 (A12). A3, A7, A12: day 3, 7, 12 of adulthood. Data were analysed using chi-square test with **p* < 0.05 and ***p* < 0.001. In addition, an example of α-synuclein-induced degeneration in the anterior DA neurons of the *C. elegans* UA44 strain, expressing both P*dat-1*::GFP + P*dat-1*::α-syn is shown (**B,C**). The DA neurons are sub-classified as four CEP neurons, which are superimposed in most pictures, and two ADE neurons. (**B**) Degeneration of CEP and ADE; (**C**) intact DA neurons.

**Table 1 ijms-21-02588-t001:** Lifespan characteristics during OLE and HT treatment.

Treatment	*n*	Mean Lifespan (days)	SEM	Days until Deaths of Population Reached				
				25%	50%	75%	90%	100%
control (DMSO)	228	22.15	0.32	18.73	22.55	25.00	26.09	34.00
Oleuropein	159	22.75	0.37	18.89	23.31	25.17	26.10	35.00
control (water)	305	20.61	0.26	17.50	19.94	23.07	25.51	34.00
Hydroxytyrosol	335	23.51**	0.26	19.64**	23.75**	25.76**	28.36**	36.00

Differences compared to control were considered significant at *p* < 0.05 (*) and *p* < 0.001 (**). *p*-value determination was realized with log-rank test for the mean lifespan and Mann–Whitney U test for specific time points.

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
