# Peer review of "Healthspan Maintenance and Prevention of Parkinson’s-like Phenotypes with Hydroxytyrosol and Oleuropein Aglycone in C. elegans"

_ijms, 2020, doi:10.3390/ijms21072588_

Round 1

Reviewer 1 Report

Brunetti et al., describe healthspan maintenance and prevention of Parkinson's syndrome after treating various PD models of C.elegans with HT and OLE. 

While the results are clear, perhaps they could have provided answers to some of the questions on possible mechanism by which this effected by testing if antioxidants would abbrogate some the beneficial effects of their test  compounds and not merely speculate.

Otherwise, I would recommend the article be published after several typographical and grammatical errors are corrected.

Author Response

We thank the reviewers for their interesting remarks that certainly made our paper more valuable. In the revised version we have taken into account the different corrections and suggestions raised by the reviewers point by point.

Comments and Suggestions for Authors
Brunetti et al., describe healthspan maintenance and prevention of Parkinson's syndrome after treating various PD models of C.elegans with HT and OLE.
While the results are clear, perhaps they could have provided answers to some of the questions on possible mechanism by which this effected by testing if antioxidants would abbrogate some the beneficial effects of their test compounds and not merely speculate.
Otherwise, I would recommend the article be published after several typographical and grammatical errors are corrected.
We re-checked the entire manuscript and corrected all detected typographical and grammatical errors.

Reviewer 2 Report

The manuscript from Brunetti et al. aims to test the effects of two olive oil polyphenols on the health- and life-span of C. elegans. Moreover, the beneficial effects of the compounds is assessed in three different C. elegans models of Parkinson’s disease. The authors reveal that both compounds can extend lifespan under stress conditions, while only one can do so under standard growth conditions. In addition, both are able to enhance the normal movement of wild-type animals and in the models of Parkinson’s. Neurodegeneration in one of the models can also be suppressed by treatment with the compounds.

This is an interesting study and I support its publication after my minor comments below have been addressed.

  1. Please be careful with terminology. elegans do not develop Parkinson’s, only humans do so. Thus, these are C. elegans models of a human disease and the worms display some phenotypes reminiscent of those seen in patients. There are numerous examples where the manuscript refers to C. elegans having Parkinson’s or displaying Parkinson’s symptoms. These all need to be amended.

Along these lines, the title needs to be amended. I would suggest “ Healthspan Maintenance and Prevention of Parkinson’s-like phenotypes in C. elegans with olive oil polyphenols”

  1. Why were the two polyphenols selected from the 36 phenolic compounds present in olive oil? The introduction provides some information, but clear justification for the choice of these two is missing.

  1. Please include images showing the pigments in untreated and treated animals in Figure 3.

  1. The movement of elegans in liquid is referred to as thrashing. I assume what is referred to as waves in the manuscript is actually thrashes. If this is the case, please use the standard ‘thrashes’ terminology.

  1. Data for animals not treated with rotenone should be included in Figure 5 to allow the rotenone-induced movement impairments to be clearly seen.

  1. Please include the wild-type data in each graph of Figure 6 and Figure 7 to allow comparisons with the OW13 and UA44 strains to be made.

  1. FUdR has been shown to affect elegans stress responses and lifespan (Van Raamsdonk and Hekimi, Mech Ageing Dev, 2011, Feldman et al., PLoS One, 2014), as well as eliciting responses after metabolism in E. coli (García-González et al., Cell, 2017). Thus, many labs have now stopped using FUdR. Please include a statement in your results to make it clear that you have used FUdR for your studies.

  1. Line 296 and 304: unc-54 needs to be in italics.

Line 468-469, and 471: all gene/transgene names need to be in italics.

Author Response

We thank the reviewers for their interesting remarks that certainly made our paper more valuable. In the revised version we have taken into account the different corrections and suggestions raised by the reviewers point by point.

Comments and Suggestions for Authors
The manuscript from Brunetti et al. aims to test the effects of two olive oil polyphenols on the health- and life-span of C. elegans. Moreover, the beneficial effects of the compounds is assessed in three different C. elegans models of Parkinson’s disease. The authors reveal that both compounds can extend lifespan under stress conditions, while only one can do so under standard growth conditions. In addition, both are able to enhance the normal movement of wild-type animals and in the models of Parkinson’s. Neurodegeneration in one of the models can also be suppressed by treatment with the compounds.
This is an interesting study and I support its publication after my minor comments below have been addressed.
Please be careful with terminology. elegans do not develop Parkinson’s, only humans do so. Thus, these are C. elegans models of a human disease and the worms display some phenotypes reminiscent of those seen in patients. There are numerous examples where the manuscript refers to C. elegans having Parkinson’s or displaying Parkinson’s symptoms. These all need to be amended.
We agree with the reviewer and changed several terms. In addition, we added one sentence for clarification in line 125/126.
Along these lines, the title needs to be amended. I would suggest “ Healthspan Maintenance and Prevention of Parkinson’s-like phenotypes in C. elegans with olive oil polyphenols”
The title was changed by using “Parkinson’s-like phenotype” instead of Parkinson’s syndrome.

Why were the two polyphenols selected from the 36 phenolic compounds present in olive oil? The introduction provides some information, but clear justification for the choice of these two is missing.
Oleuropein, the more abundant olive leaf polyphenol, belongs to a specific group of coumarin-like compounds, the secoiridoids, which are abundant in Oleaceae. Secoiridoids are compounds usually bound to glycosides in the plant, where they arise from the secondary metabolism of terpenes. The secoiridoids, found only in plants belonging to the family of Oleaceae, which includes Olea europaea L., are characterized by the presence of elenolic acid in their molecular structure. In particular, oleuropein and ligstroside structures contain the phenyl ethyl alcohol (hydroxytyrosol and tyrosol, respectively) esterified with elenolic acid. Accordingly, oleuropein is the ester of hydroxytyrosol (3,4-DHPEA) and elenolic acid (EA) glucoside (an oleosidic skeleton common to the secoiridoid glucosides of Oleaceae) (Polyphenols in Olive Oil Antonio Segura-Carretero, ... Alberto Fernández-Gutiérrez, in Olives and Olive Oil in Health and Disease Prevention, 2010).
Several studies have provided data that, among others, indicate that:
 Oleuropein aglycone treatment throughout cellular lifespan delays senescence and maintains the young cellular morphology for longer.
 Oleuropein aglycone treatment results in reduced cellular oxidative cargo and lower levels of intracellular ROS.
 Oleuropein aglycone treatment attenuates the decrease of proteasome activities during the senescence process.
 Oleuropein aglycone treatment increases the overall proteolysis rates following oxidative stress.
 Oleuropein aglycone treatment confers higher cellular survival rates following oxidative stress.
 Oleuropein aglycone reduces the inflammatory response in everal cellular and tissue models
 Oleuropein aglycone regulates cell metabolism similarly to caloric restriction via activation of AMPK and reduction of the mTORC activity
 Oleuropein aglycone is a natural anti-aging compound that offers the advantage that it can be easily taken up through normal diet.
Interestingly, hydroxytyrosol plasma concentration does not only derive from olive oil polyphenols absorption, but also from dopamine degradation. In fact, hydroxytyrosol is a by-product of dopamine oxidative metabolism, a pathway that involves multiple enzymes, monoaminooxidase, aldehyde dehydrogenase and aldehyde reductase. It resulted be interestingly considering that Parkinson’s disease (PD) is characterized by the progressive loss of dopaminergic neurons in the midbrain region. Current therapies for PD are not able to prevent dopaminergic neuron loss or stop the progression of the disease, only delay the onset or reduce the motor symptoms. The gold standard therapy against PD relies on restoring the optimum level of dopamine (DA) and its associated signaling pathways by the administration of L-3,4-dihydroxyphenylalanine (L-DOPA), a precursor of DA.
Analyzing all these foundamental aspects we decided to use OleA and its main metabolite HT in this study in order to validate and to optimize their use for possible PD prevention and therapy.

Please include images showing the pigments in untreated and treated animals in Figure 3.
We added some representative example pictures to Figure 3 (now Figure 4) and referred to them in the legend.

The movement of elegans in liquid is referred to as thrashing. I assume what is referred to as waves in the manuscript is actually thrashes. If this is the case, please use the standard ‘thrashes’ terminology.
We replaced the “waves/min” and “wave initiation rate” terms by “thrashes/min” and “thrashing rate”, respectively, throughout the manuscript.

Data for animals not treated with rotenone should be included in Figure 5 to allow the rotenone-induced movement impairments to be clearly seen.
We added the data for rotenone-untreated animals to figure 5 (now Figure 6).

Please include the wild-type data in each graph of Figure 6 and Figure 7 to allow comparisons with the OW13 and UA44 strains to be made.
We added the wild type data for untreated animals to Figure 6 and 7 (now Figure 7 and 8).

FUdR has been shown to affect elegans stress responses and lifespan (Van Raamsdonk and Hekimi, Mech Ageing Dev, 2011, Feldman et al., PLoS One, 2014), as well as eliciting responses after metabolism in E. coli (García-González et al., Cell, 2017). Thus, many labs have now stopped using FUdR. Please include a statement in your results to make it clear that you have used FUdR for your studies.
We agree that the usage of FUdR is criticisable. In line 430-434, we discussed that the usage of FUdR could potentially produce misleading results.

Line 296 and 304: unc-54 needs to be in italics.
We changed that as suggested.

Line 468-469, and 471: all gene/transgene names need to be in italics.
The gene names in the transgene descriptions are italics now.

Reviewer 3 Report

The manuscript of Brunetti et al. describes the protective properties of hydroxytyrosol and oleuropein aglycone in C. elegans.

The manuscript is well-written and its scientific message is interesting, considering the continuous interest of the scientific community in studying the role of nutraceuticals as interesting compounds for human health.

Nevertheless, the manuscript must be improved both in term of formatting and content explanation.

A chemical structure of the molecules used as protectors could be useful.

The discussion about the role of the compounds in health and lifespan is not so strong. The results miss of some molecular information that can help to explain the role of the compounds (antioxidant, antiinflammatory, other mechanisms). The manuscript is a description of functional tests, but a molecular analysis is mandatory to explain the mechanisms of action.

I kindly suggested to improve the quality of the pictures: you have three graphs for a picture, so I suggest to use the blank to add some information like the legend, short title, so the pictures will be self-explanatory.

Could the authors explain the reason for the use of heat stress to test the compounds? Is it a common test used for nematodes? PD is a different condition.

Please, improved the description of Figure 1: add parenthesis to the concentration as n (30 ug/mL) = 74, to improve readability. Remove useless parenthesis line 145.

What is the meaning of the statistical analysis in Figure 1? Usually, the analysis should be performed every day and not as an "average".

Also, a statistical analysis could be helpful to correctly evaluate the real difference between the survival rate of 15 and 22% of 250 and 500 mg of OLE.

It's interesting that in OLE-treated nematodes, 500 dose seems to be more effective, but in 250 the nematodes live longer. Also, in HT-treated nematodes, it's interesting to observe that the dose-effect is lost at 500 doses.

Figure 1: please, consider to add the name of the compounds to the pictures, so they can be easily read and understood.

Figure 4: I suggest to add a short title of each graph: the pictures should be self-explanatory.

Figure 5: the legends in the pictures don't show to the reader that the nematodes were treated with rotenone.

Could the authors, compare the amount of compounds used with the amount of compounds in the human diet?

Author Response

We thank the reviewers for their interesting remarks that certainly made our paper more valuable. In the revised version we have taken into account the different corrections and suggestions raised by the reviewers point by point.

Comments and Suggestions for Authors

The manuscript of Brunetti et al. describes the protective properties of hydroxytyrosol and oleuropein aglycone in C. elegans. The manuscript is well-written and its scientific message is interesting, considering the continuous interest of the scientific community in studying the role of nutraceuticals as interesting compounds for human health.

Nevertheless, the manuscript must be improved both in term of formatting and content explanation.

A chemical structure of the molecules used as protectors could be useful.

We added the chemical structures in the new Figure 1.

The discussion about the role of the compounds in health and lifespan is not so strong. The results miss of some molecular information that can help to explain the role of the compounds (antioxidant, antiinflammatory, other mechanisms). The manuscript is a description of functional tests, but a molecular analysis is mandatory to explain the mechanisms of action.

I kindly suggested to improve the quality of the pictures: you have three graphs for a picture, so I suggest to use the blank to add some information like the legend, short title, so the pictures will be self-explanatory.

Thank you very much for this good suggestion. We added a more detailed legend to this free space in the swim assay figures and some short titles for each graph. Thus, the pictures should be self-explanatory now.

Could the authors explain the reason for the use of heat stress to test the compounds? Is it a common test used for nematodes? PD is a different condition.

We added an explanation in lines 132-135.

Please, improved the description of Figure 1: add parenthesis to the concentration as n (30 ug/mL) = 74, to improve readability. Remove useless parenthesis line 145.

We changed the figure legend as suggested.

What is the meaning of the statistical analysis in Figure 1? Usually, the analysis should be performed every day and not as an "average".

For the heat stress survival analysis as well as for the lifespan assay, the log-rank test was used to check for statistical differences. This test compares the survival distributions of two samples and significances are given for the whole curve instead for single days.

Also, a statistical analysis could be helpful to correctly evaluate the real difference between the survival rate of 15 and 22% of 250 and 500 mg of OLE. It's interesting that in OLE-treated nematodes, 500 dose seems to be more effective, but in 250 the nematodes live longer. Also, in HT-treated nematodes, it's interesting to observe that the dose-effect is lost at 500 doses.

The log-rank analysis was repeated and the treatments were compared with each other. However, no significant difference was found between the two highest concentrations. This finding was added to the results (lines 139-147)

Figure 1: please, consider to add the name of the compounds to the pictures, so they can be easily read and understood.

We added the names of the test-substances on the head of the graphs.

Figure 4: I suggest to add a short title of each graph: the pictures should be self-explanatory.

Each swim assay graph is now applied with a short title and a more detailed legend.

Figure 5: the legends in the pictures don't show to the reader that the nematodes were treated with rotenone.

The legend for Figure 5 (which is now Figure 6) as well as for all swim assay graphs was edited. Now the conditions of the assay are easier to understand.

Could the authors, compare the amount of compounds used with the amount of compounds in the human diet?
Polyphenols concentration in EVOO depends on several variables such as (i) the olive cultivar and the ripening stage of fruit; (ii) environmental factors (altitude, cultivation practices, and amount of irrigation); (iii) extraction conditions (heating, added water and malaxation); (iv) extraction systems used to separate oil from olive pastes (pressure, centrifugation systems); and (v) storage conditions and time, due to spontaneous oxidation, and suspended particle deposition (https://doi.org/10.1002/1438-9312(200210)104:9/10). At the best, the content of OLE in EVOO can reach levels exceeding 60 mg/100 g. In addition, data about the bioavailability of olive polyphenols, notably oleuropein and hydroxytyrosol, are scarce due to several factors hindering their absorption, tissue distribution and intracellular penetration. These include intrinsically reduced absorption, gut microbiota metabolism, extensive modifications in liver favouring renal excretion of their metabolites. HT is well absorbed in the gastrointestinal tract in a dose-dependent manner but its bioavailability is poor due to an important pre-systemic metabolism both in gut and liver, leading to the formation of sulfate and glucuronide conjugates, such that its concentrations in body fluids in free form (free HT) were deemed undetectable so far.
This fact leads to the controversy whether HT concentrations in human tissues after olive oil ingestion are too low to explain the observed biological activities. However, some authors measured HT levels in plasma following oral administration or EVOO intake (https://doi.org/10.1021/ac000121h), and, in November 2011, the European Food Safety Authority (EFSA) released a health claim concerning the benefits of the ingestion of 5 mg/day phenolic compounds from olive oil (including HT, tyrosol, and their secoiridoids) for LDL protection against oxidation (http://dx.doi.org/10.2903/j.efsa.2011.2033).
Accurate studies on the effective daily dose of olive polyphenols to be administered to humans to get significant protection are still lacking, and it must be taken into account that, apparently, the amount of OleA and other plant polyphenols present in foods is not adequate to ensure daily doses suitable to get short-term acute protection. However, clinical and experimental evidence suggests that the continuous assumption of foods containing moderate amounts of these molecules can be effective in the long term, also due to their possible accumulation as lipophilic molecules, producing a low-intensity continuative hormetic stimulus improving cell defences against oxidative stress, amyloid deposition and other alterations underlying aging-associated pathologies. Nevertheless, the low daily consumption of olive oil polyphenols with a typical Mediterranean Diet suggests the value of the integration of polyphenol-enriched olive leaf extracts that can intensify, in the short-term, the beneficial effects of these molecules.
Anyway, data demostrated that, even if the blood brain barrier (BBB) has an extremely selective permeability, the aglycones of polyphenols can cross membranes by a passive diffusion mechanism.

Indeed, it was demostrated that Oleuropein aglycone and Hydroxytyrosol can cross the BBB, an essential feature to explain its neuroprotective role.

Round 2

Reviewer 3 Report

The revised manuscript improved significantly the quality of the presentation, the understandability and readiness.